# Associations Between Care Environments and Environmental Modifications in the Daily Living Settings of Children with Medical Complexity

**DOI:** 10.3390/nursrep15110400

**Published:** 2025-11-13

**Authors:** Yumi Mizuochi, Yukako Shigematsu, Yoshitomo Fukuura

**Affiliations:** 1Doctoral Program, Graduate School of Medicine, Kurume University, 777-1 Higashikushiharamachi, Kurume-shi 830-0003, Fukuoka, Japan; 2Department of Nursing, School of Medicine, Kurume University, 777-1 Higashikushiharamachi, Kurume-shi 830-0003, Fukuoka, Japan; shigematsu_yukako@kurume-u.ac.jp (Y.S.); fukuura_yoshitomo@kurume-u.ac.jp (Y.F.)

**Keywords:** children with medical complexity, family, professional, environment, environmental modification, daily living settings

## Abstract

**Background/Objectives:** Children with medical complexity (CMC) living in community settings are increasingly prevalent, and appropriate care environments are essential to support their wellbeing. This study aimed to examine the relationship between care environments and environmental modifications in CMC’s daily living, as well as the factors influencing this relationship. **Methods:** A cross-sectional survey was conducted among families of CMC and professionals (including visiting nurses, consultation support specialists, and other professionals) across Japan. Mann–Whitney U tests were used to examine differences between families and professionals, as well as by the agent of modification. Wilcoxon signed-rank tests were performed to compare environmental scores before and after modifications. Spearman’s rank correlation coefficients were used to assess associations between environmental modification scores and post-modification environmental scores. Multiple regression analyses were conducted to identify predictors of post-modification environmental scores, including environmental modification subcategory scores and background characteristics. **Results:** Valid responses were obtained from 90 families (93.8%) and 221 professionals (76.2%). Significant differences in environmental modification scores were observed between families and professionals. Scores were significantly higher when modifications were conducted jointly by families and professionals and when led by professionals than when implemented solely by families. Pre- and post-modification environmental scores demonstrated significant improvements across all domains—physical, collaborative, service, and community—as well as in total score. **Conclusions:** Families may underappreciate professional psychological support as part of environmental modifications. Jointly planned, family-centred modifications integrating physical, service, collaborative, and community elements improve care environments and support the wellbeing of CMC and their families.

## 1. Introduction

Children with medical complexity (CMC) are children with chronic conditions that affect multiple organ systems and require ongoing medical care and support, often including reliance on medical technology. Recently, remarkable advances in perinatal medicine have made it possible for clinicians to save extremely premature infants, resulting in an increasing number of CMC living at home and in community settings [1,2,3,4]. These children often experience limitations in activities of daily living, require coordinated, multispecialty care, and frequently utilize healthcare services [5,6]. Although they vary in developmental stage, physical condition, and medical care needs across the pediatric age range from infancy through adolescence [6,7], they continue to grow and develop regardless of the severity of their disability. The care environment shapes their daily living conditions and is instrumental in promoting their growth and development [8,9].

For CMC to live safely and comfortably, their environment must be comprehensively structured to include a physical environment that supports daily care, collaborative environment in which their families take the lead in cooperating with professionals, service environment that ensures access to necessary resources, and community environment that accepts and supports them and their families [10]. When these elements are in place, CMC and their families can confidently engage in ordinary outings [11], with the children interacting with their peers, while their families connect with community members and participate in the society [12]. Establishing an appropriate care environment in daily living settings allows CMC growth and healthy development within the families and communities of these children [13]. Moreover, it supports the active engagement of their families in daily life and social activities [14], ultimately contributing to the wellbeing of both CMC and their families.

Therefore, environmental modifications are essential for promoting the healthy growth and development of CMC, reducing the care burden on families, and creating living settings where care can be provided safely and calmly. In daily living settings without well-structured collaborative and service environments, it is difficult for CMC to receive care in a home-like environment that promotes developmental and physiological stability [15]. For families, care often involves night-time responsibilities such as suctioning and repositioning, and inadequate support for the family care environment can lead to sleep deprivation, chronic fatigue, and exhaustion, ultimately reducing quality of life [16]. Moreover, in the absence of a strong collaborative environment, families may face physical strain and health problems related to chronic illness, psychological stress, and psychosocial challenges such as isolation and helplessness [17,18,19]. In addition, without supportive community environments, families raising CMC are likely to experience social stigmatization and exclusion, weakened community ties, and barriers to social participation [20]. Families caring for CMC are known to encounter physical, psychological, emotional, and financial stressors [21,22], many of which can be alleviated through appropriate environmental modifications in daily living settings [23]. Healthcare professionals must recognize families as the central agents of care and, in partnering with them [24], provide evidence-based information [25], facilitate access to resources, coordinate with multidisciplinary teams and community networks [26], and offer psychosocial support [27] to empower them in autonomously adapting their care environments.

In hospitals, healthcare professionals primarily lead environmental modifications; however, in daily living settings, families are often the primary agents of such modifications [28]. As family members are familiar with the physical environment of the home and broader living context of all family members, they actively implement environmental modifications that integrate medical care into everyday life. Therefore, in daily living settings, families need to take the lead while collaborating with professionals to establish supportive environments. Research on environmental modifications for CMC has highlighted challenges such as discharge preparation and care planning that do not incorporate the knowledge and perspectives of family members [29], difficulties caused by complex structures within medical and welfare systems, lack of coordination across institutions, gaps in intersystem communication, and barriers to care continuity [14,21]. However, few studies have focused on the environments surrounding CMC and their families or on the process of environmental modification.

Accordingly, the purpose of this study was to clarify the relationship between care environments and environmental modifications in the daily living settings of CMC, identify influencing factors, and explore strategies for environmental modification that enhance the wellbeing of CMC and their families.

## 2. Materials and Methods

### 2.1. Study Design

This was a cross-sectional observational study that employed a self-administered questionnaire.

### 2.2. Ethical Considerations

This study was approved by the Kurume University Ethics Committee (approval number: 24109).

Written explanations of the survey, such as the overview, methods, ethical considerations for participants, and dissemination of results, were presented to facility representatives. Questionnaires and request forms were then distributed to the participants by the facility representatives.

To ensure anonymity, the questionnaires were not signed and did not contain identifying information. Respondents were informed that participation was voluntary, refusal to participate would not result in any disadvantages, and consent would be confirmed by checking the consent box at the beginning of the questionnaire. They were also informed that results would be published and that because the data were pseudonymized, questionnaires could not be withdrawn or deleted after submission.

### 2.3. Definition of Terms

Daily living settings: places where children requiring medical care live with their families.

Care environment: Environment surrounding daily life and care, consisting of:Physical environment: arrangement of medical equipment, beds, and supplies in the child’s living space.Collaborative environment: collaboration between families and professionals to support CMC and their families.Community environment: community in which the children live with their families.Service environment: services that support the daily lives of the children and their families.

Environmental modifications: Activities to structure or adjust the environment of the children and their families in daily living settings. These include activities undertaken by their families alone, those carried out jointly by their families and professionals, and those led by professionals.

Physical environmental modifications: modifications to improve the physical environment.Family-led modifications: Modifications implemented by a collaboration of families and professionals to respect family autonomy.Family-led, facilitated by the professional role, environmental modifications: Modifications implemented by professionals based on their specialized expertise.Service environmental modifications: Modifications that organize service environments according to the needs of CMC and their families.Community environmental modifications: Modifications that develop community environments that accept and support CMC and their families.Care improvement modifications: Modifications that clarify and address challenges in service provision, led by professionals in collaboration with families.

Family: Parents, grandparents, and siblings of the child (including co-resident and non-co-resident family members). Professionals: medical and welfare professionals involved in environmental modifications in daily living settings.

### 2.4. Sample Size Determination

Multiple regression analyses were conducted to examine the relationships between environmental modifications (48 items) and care environments (35 items). Because both families and professionals were involved in environmental modifications for CMC, both groups were included as participants in this study.

The required sample size was estimated based on Cohen’s (1988) criterion (N ≥ 50 + 8 m) [30], assuming 10 explanatory variables, indicating that at least 130 participants were needed. Additionally, a power analysis using G*Power 3.1, with a medium effect size (f^2^ = 0.15), a significance level of 0.05, and a power of 0.80, yielded a required sample size of approximately 118 participants. The required sample size refers to the total number of participants, including both families and professionals, necessary for the multiple regression analyses.

Ultimately, 311 valid responses were obtained, including 90 family questionnaires and 221 professional questionnaires, which exceeds the required sample size and satisfies the power analysis requirements.

### 2.5. Participation and Data Collection

Study participants were families and professionals with experience in environmental modifications within the daily living settings of CMC. Professionals included visiting nurses, consultation support specialists, and nurses working at child development support centres and after-school day service centres.

From September 2024 to March 2025, prefectures across Japan were selected based on regional and population characteristics. A simple random sample was drawn, comprising 500 visiting nurse stations providing home care support for CMC, 200 care management offices, and 150 child development support and after-school day service centres attended by CMC. Additionally, the researchers requested cooperation from 26 family associations of CMC.

Researchers formally requested in writing that facility administrators and representatives of family associations distribute study information sheets and questionnaires to families and professionals with experience in environmental modifications for CMC. The information sheets clearly described the study objectives, assurance of anonymity, voluntary nature of participation, and the right to refuse to answer. Researchers did not provide any oral explanations or direct contact with potential participants. Participation was entirely voluntary and based on the participants’ free will. Responses were returned individually by mail. A total of 2200 professional questionnaires and 1000 family questionnaires were distributed among the participating facilities.

Questionnaires were distributed by mail. Based on the estimated number of staff involved with CMC, each visiting nurse station received one family questionnaire and three professional questionnaires; each care management office received one family questionnaire and two professional questionnaires; and each child development support or after-school day service centre received two family questionnaires and two professional questionnaires.

Furthermore, at the request of participating family associations, the family questionnaire was also made available both as a mail-in form and as a web-based survey (Google Forms). Families and professionals without experience in environmental modifications for CMC were excluded.

### 2.6. Instruments

#### Questionnaire Items

Characteristics of families and professionals

For families, basic attributes included the primary caregiver, presence and number of siblings, employment status of the primary caregiver, presence of interactions with other family members, presence of a person to consult with, and family relationships. For professionals, attributes included professional role during environmental modifications, presence of other institutions or professionals for consultation, relationships among professionals, experience with service coordination, and participation in training sessions or study groups. Environmental modification case items included the context of environmental modification, child’s health status, child’s signs and responses, child’s expression of intent, and background of environmental modifications (availability of desired services, ease of service use, information about needed services, people the family communicates with, people the family can ask for help, and people with whom the family can share private matters).

2.Items related to the care environment

Items related to the care environment were extracted from the conceptual framework in a previous review [10]. Thirty-five items asked about the environment pre- and post-environmental modifications. Responses were rated on a 7-point Likert scale: ‘very well organized (6)’, ‘well organized (5)’, ‘somewhat organized (4)’, ‘neither well nor poorly organized (3),’ ‘not very well organized (2),’ ‘hardly organized (1),’ and ‘not organized at all (0)’. In addition, for aspects unrelated to environmental modifications, the option ‘not applicable’ was provided, and responses were treated as missing values when selected. The complete list of items is provided in Appendix A.

3.Items related to environmental modifications

Items related to environmental modifications were extracted from interviews with 20 professionals involved in environmental modifications in the daily living settings of CMC, including hospital nurses, home-visit nurses, consultation support specialists, nurses working in child development support and after-school day services, medical social workers, and public health nurses. Overall, 48 items assessed the extent of environmental modifications implemented. Responses were rated on a 5-point Likert scale: ‘implemented extensively (5)’, ‘implemented (4)’, ‘somewhat implemented (3)’, ‘rarely implemented (2)’, and ‘not implemented (1)’. Additional options of ‘not necessary’ and ‘don’t know’ were provided, and responses were treated as missing values when selected. Of the 48 items, 19 were related to modifications performed by families, 9 were related to modifications performed jointly by families and professionals, and 20 were related to modifications performed by professionals. The complete list of items is provided in Appendix A.

### 2.7. Data Analysis

Statistical analyses were conducted using IBM SPSS Statistics, version 30 (IBM Corp., Armonk, NY, USA). For each analysis, participants with missing data on any of the variables included were excluded from that analysis, and only complete cases were used. A *p*-value of less than 0.05 was considered statistically significant for all analyses, and all tests were two-sided.

#### 2.7.1. Analysis of Differences in Environmental Modification Scores Between Families and Professionals and by Agent of Modification

To examine differences in environmental modification scores between families and professionals, as well as by agent of environmental modification, the normality of the data was first assessed using the Shapiro–Wilk test. Since the assumption of normality was not met, differences were analyzed using the Mann–Whitney U test.

#### 2.7.2. Comparison of Care Environment Scores Pre- and Post-Environmental Modifications

For each item, the normality of pre- and post-modification care environment scores was assessed using the Shapiro–Wilk test. As the normality assumption was not met, differences in care environment scores pre- and post-environmental modifications were analyzed using the non-parametric Wilcoxon signed-rank test. This test is appropriate for evaluating changes in the median of paired ordinal data. Median and interquartile range (IQR) values are reported to describe the distribution of pre- and post-modification scores.

#### 2.7.3. Analysis of Associations Between Environmental Modification Scores and Post-Modification Care Environment Scores

Spearman’s rank correlation coefficient (*ρ*) was used to assess the relationship between environmental modification scores and post-modification care environment scores.

#### 2.7.4. Analysis of Associations Between Post-Modification Care Environment Scores and Environmental Modification Scores by Category

To examine which environmental modification subcategories contribute to the post-modification care environment, multiple regression analysis using the forced entry method was conducted. The total scores of each post-modification care environment (physical environment, collaborative environment, community environment, and service environment) served as dependent variables, while the total scores of each environmental modification subcategory (physical modifications, family-led modifications, facilitated by the professional role, community modifications, service modifications, and care improvement modifications) were included as independent variables.

#### 2.7.5. Analysis of Background Factors Influencing the Care Environment Post-Modifications

Multiple regression analysis using the forced entry method was conducted with the total score of the post-modification care environment as the dependent variable and background factors of environmental modifications as independent variables.

## 3. Results

### 3.1. Sample Description Descriptive Statistics

Of the distributed questionnaires, 68 family questionnaires were returned by mail (response rate: 6.8%), and an additional 28 responses were obtained online from 17 family associations. For professionals, 290 questionnaires were returned (response rate: 13.2%). After excluding ineligible or incomplete responses, 90 family questionnaires (number of valid responses/number of questionnaires collected = 93.8%) and 221 professional questionnaires (number of valid responses/number of questionnaires collected = 76.2%) were included in the final analysis, yielding a total of 311 valid responses.

The basic characteristics of the families and professionals, as well as the attributes of the environmental modification contexts, are summarized in Appendix A.

### 3.2. Analysis of Differences in Environmental Modification Scores Between Families and Professionals, as Well as by Agent of Modification

#### 3.2.1. Differences in Environmental Modification Scores Between Families and Professionals

Significant differences were observed between families and professionals in the total environmental modification score, physical modifications, family-led modifications facilitated by professionals, community modifications, service modifications, and care improvement environmental modifications.

#### 3.2.2. Differences in Environmental Modification Scores Between Families and Professionals, and by Agent of Modification

No significant differences were found for modifications conducted solely by families. By contrast, significant differences were observed for modifications conducted jointly by families and professionals and for those conducted by professionals alone.

Detailed results, including all statistical values, are provided in Appendix A.

### 3.3. Differences in Care Environment Scores Pre- and Post-Environmental Modifications

Significant improvements were observed in care environment scores in daily living settings following environmental modifications. Improvements were noted in the physical, collaborative, service, community, and overall care environment domains. Detailed results, including all statistical values, are provided in Appendix A.

### 3.4. Correlations Between Environmental Modification Scores and Post-Modification Care Environment Scores

Environmental modification scores showed significant moderate positive correlations with post-modification care environment scores. Among these, the strongest association was found between environmental modification scores and post-modification collaborative environment (*ρ* = 0.504, *p* < 0.001) (Table 1).

### 3.5. Associations Between Post-Modification Care Environment Scores and Environmental Modification Scores

Multiple regression analyses were conducted to examine the associations between environmental modification subcategories and post-modification care environment scores. The results are summarized below (full regression table provided in Appendix A).

Physical environment: No significant associations were observed for any environmental modification subcategory.Collaborative environment: Significant positive associations were found for physical modifications (β = 0.396, *p* = 0.025) and family-led modifications (β = 0.331, *p* = 0.006). Other modifications were not significantly associated.Community environment: Care improvement modifications showed a significant positive association (β = 0.603, *p* = 0.001), while other modifications were not significant.Service environment: Significant positive associations were observed for family-led modifications (β = 0.305, *p* = 0.006) and care improvement modifications (β = 0.686, *p* < 0.001). No other modifications were significant.

### 3.6. Background Factors Influencing the Care Environment Post-Environmental Modifications

Among background factors, service accessibility had a significant positive effect on the care environment post-environmental modifications (β = 0.286, *p* = 0.020). In contrast, availability of desired services (β = −0.047, *p* = 0.686), information about necessary services (β = −0.166, *p* = 0.162), presence of relatives, friends, or acquaintances with whom families could communicate (β = 0.090, *p* = 0.452), presence of relatives or acquaintances from whom families can seek help (β = −0.027, *p* = 0.845), presence of relatives or acquaintances with whom families discuss private matters (β = 0.147, *p* = 0.292), and relationship with professionals (β = 0.095, *p* = 0.356) were not significantly associated with post-modification care environment scores (Table 2).

## 4. Discussion

This study found that significant improvements in care environment scores were observed following the environmental modifications. In addition, for nearly all environmental modification items, professionals reported higher environmental modification scores than families. Modifications conducted by families and professionals, as well as those conducted by professionals alone, scored significantly higher than those for families alone.

Therefore, families and professionals differ in their perceptions of the environmental modifications implemented by professionals. Professionals reported that, as part of environmental modifications, they provided information and psychological support to help families feel empowered and make decisions. However, families may not have perceived these activities as environmental modifications, as they may not have seemed intentional. As professionals tend to adopt a comprehensive perspective on the environment [31], they also provide support for less visible aspects of environmental modification, such as collaboration, community work, and service. Thus, this lack of visibility likely influenced the families’ perceptions and awareness. In addition, service accessibility was identified as a factor influencing environmental modifications. For families raising CMC, a lack of essential medical and welfare services, as well as insufficient resources such as medical equipment and appropriate housing, may compromise the child’s health and the quality of life of both the child and the family [32]. Therefore, the ease of access to services and resources was an important factor directly affecting the implementation and effectiveness of environmental modifications. In other words, service accessibility should not be regarded merely as a matter of convenience, but rather as a foundational condition for achieving family-centred environmental modifications.

Previous studies have also reported gaps in modification recognition between families and professionals. Woldring et al. identified discrepancies in the perceptions of family involvement in care between families and healthcare providers [33]. In addition, barriers such as families’ lack of understanding and mismatched expectations in family-centred interventions have also been reported [34]. Such gaps in modification recognition can affect trust-building and hinder collaborative activities between families and professionals [35].

Therefore, it is essential to make environmental modifications more visible and enlighten families about the structure of the care environment in daily living settings, which will result in environments tailored to reflect the diversity of CMC. To achieve this, evaluation frameworks and indicators that allow both families and professionals to understand the progress and outcomes of environmental modifications must be developed. Such efforts may reduce discrepancies in modification recognition between families and professionals, improving the quality of life of CMC and their families.

This study determined that environmental modifications generally improve the care environment in the daily living settings of CMC.

Significant increases in service environment scores were observed following the environmental modifications. As the wellbeing of CMC and their families improves when the service environment is adequately structured [36,37], particular emphasis should be placed on this domain. Prior research has emphasized that enhancing the quality of the service environment for CMC requires collaborative coordination between local care resources and specialized medical institutions [38]. However, this study revealed that families often played a central role in service-related modifications. Improvements in the service environment were associated with both collaborative modifications conducted by families and professionals, and service-related modifications. Therefore optimizing the service environment requires expansion of services provided by professionals and active collaboration in which families take the lead with the backing of professionals. When families actively engage in service coordination, the service environment improves and local care resources are strengthened. In other words, even if services are available and interprofessional collaboration is promoted, the service environment cannot be adequately structured without families’ proactive utilization of services. Thus, collaboration between families and professionals facilitates smoother service coordination [39]. Enhancing family autonomy and empowerment enables professionals to design services that reflect the values and everyday lives of CMC and their families.

A distinctive feature of service environment modifications in the daily living settings of CMC is the tendency toward fragmented care, owing to the need to access multiple health and welfare services. Without integrated environmental modifications led by the professionals involved, such fragmentation may create confusion [40]. Active family participation in service introduction and coordination, together with professional collaboration, is necessary to ensure that the service environment meets the needs of both CMC and their families. The results of this study support the importance of such collaborative environmental modifications [38,41].

This study also found that ‘service accessibility’ was an influencing factor for environmental modifications. Effective modifications require specific resources. For households with CMC, essential resources include medical equipment and home adjustments, as resource shortages can lead to deteriorations in child health and quality of life [42]. Our findings indicate the importance of resource availability and accessibility. As CMC are in a period of rapid development, more frequent adjustments and readjustments are needed than those required for adults [43]. Proactive information sharing and systems that facilitate easy access to services are essential for preparing in advance for developmental changes. In addition, having accessible contact points and care coordinators enables families to consolidate information, streamline service coordination, and reduce burden. A single point of contact with clear roles and responsibilities is recommended [44]. Simplified procedures and timely service implementation also reduce family burden and promote service utilization [45,46]. Considering the various health and welfare professionals involved in caring for CMC [26,40], systems such as region-specific ICT records that enable information sharing on family care plans and consultation histories across disciplines, as well as regular family-inclusive case conferences, are needed. Furthermore, institutional designs by local governments and administrative bodies are required to ensure smooth implementation of services.

Physical environment scores significantly improved post-environmental modifications, and an association was found between the overall environmental modification score and post-modification physical environment score. However, no associations were found between individual subcategories of environmental modifications and the post-modification physical environment. This may be due to the physical environment encompassing elements of the home such as renovations, placement of medical equipment, and adjustments to daily living flow, all of which shape the living conditions of CMC and their families. Previous studies have reported that physical barriers within the home, such as stairs and bathrooms, limits the daily lives and care of CMC, and that small-scale renovations or equipment adjustments alone are insufficient to improve quality of life or promote social participation [47]. In daily living settings, the physical environment, as well as collaborative, service, and community environments, must be developed [48] to establish care environments that facilitate child care, support family caregiving, and promote outings for CMC and their families. Briefly, improvements to the physical environment alone do not directly lead to overall quality-of-life gains; environmental modifications must also consider the broader context of family life within the community. Specifically, structuring physical environments that enable CMC and their families to go out more easily facilitates opportunities for peer interaction, participation in community events and peer group activities, access to schooling for children, and employment for family members. Thus, both mobility-related modifications and the integration of collaborative, service and community environments are required. Such physical environment modifications enable greater participation of CMC and their families in community life, reduces social isolation, and improves wellbeing.

Scores for the collaborative environment significantly improved following environmental modifications. Improvements in the collaborative environment were associated with physical environmental and family-led collaborative modifications. When families take the lead and collaborate with professionals to structure the physical environment, they ensure the sustained availability of the medical care CMC require and maintain the comfort of the home as a living space [49]. Joint family and professional efforts to modify the physical environment strengthen family-centred care and foster collaborative relationships between families and professionals [50]. Physical modifications implemented with attention to the collaborative environment promote family empowerment [38] and contribute to the realization of the autonomy of daily family life. To advance collaboration between families and professionals, it is essential for families to share information about the child’s care environment and family’s living conditions with professionals, and for both groups to jointly consider what constitutes a comfortable and supportive environment. This shared understanding provides the foundation for building a truly collaborative care environment.

Scores for the community environment also improved following environmental modifications. Improvements in the community environment were associated with collaborative modifications and professional-led service modifications. Thus, incorporating services and social resources may help create community environments that positively influence the overall lives of families. Professionals, through the introduction and coordination of services, act as a bridge between CMC and their families and communities.

When CMC and their families are accepted by the community in which they live through repeated interactions with local residents, they are more likely to receive informal support in emergencies such as natural disasters [51]. To foster such relationships, professionals can encourage families to participate in community events and engage with local volunteers to gain visibility and build connections. Moreover, active family involvement in community activities geared toward disaster-preparedness, such as evacuation drills, allows the broader community to be involved in supporting families in raising CMC in times of crisis. Such activities deepen community members’ understanding of the daily lives and support needs of CMC. Through these initiatives, families of CMC can be more accepted in their communities, resulting in community environments where they can live with greater security in everyday life and during emergencies.

This study has some limitations. First, as the data collected in this study were based on participants’ recalled experiences of environmental modifications, the accuracy of memory and potential influence of recall bias represent an important limitation. Second, it was conducted at a limited number of facilities in Japan, and further studies in other countries and regions are needed to examine the applicability of the findings across different healthcare systems and support structures. Third, the participants were limited to professionals and families, without including all relevant stakeholders such as administrative officials and public health nurses involved in policy-making for environmental modification. Informal community resources that provide support were also not considered. Future research should therefore broaden the scope of participants. Fourth, as this study employed a cross-sectional design and relied on self-reported data, its ability to clarify causal relationships and generalize findings is limited. Future studies should seek to replicate the findings with larger samples, and employ longitudinal and interventional designs to examine the long-term effects of family-centred environmental modifications. Fifth, although the valid response rate in this study was high, the overall collection rate was relatively low, which limits the generalizability of the results. It is necessary to consider the possibility of selection bias due to differences in characteristics between respondents and non-respondents. Despite these limitations, the present study contributes foundational insights by clarifying the relationship between care environments and environmental modifications in the daily living settings of CMC, identifying influencing factors, and providing a basis for exploring strategies to enhance the wellbeing of CMC and their families.

## 5. Conclusions

This study revealed differences in recognition regarding environmental modifications, suggesting that families may not perceive the psychological support provided by professionals as part of environmental modifications.

Environmental modifications in the daily living settings of CMC led to improvements across all domains of the care environment: service, physical, collaborative, and community. Collaborative environments were associated with family-led and physical modifications, community environments with care improvement modifications, and service environments with both care improvement and family-led modifications.

These findings underscore the importance of a family-centred, collaborative approach, demonstrating that, when families and professionals jointly plan and implement environmental modifications, the care environment in daily living settings for CMC can be improved. Moving forward, environmental modifications must actively promote family participation and integrate physical, service, collaborative, and community dimensions in order to optimize care environments and enhance the wellbeing of CMC and their families.

## Figures and Tables

**Table 1 nursrep-15-00400-t001:** Associations between Environmental Modification Scores and the Post-modification Care Environment.

Variable 1	Variable 2	Correlation Coefficient (*ρ*)	*p* Value	Sample Size (n)
Total environmental modification score	Post-modification care environment	0.497 **	<0.001	117
Total environmental modification score	Post-modification physical environment	0.465 **	<0.001	135
Total environmental modification score	Post-modification collaborative environment	0.504 **	<0.001	125
Total environmental modification score	Post-modification service environment	0.499 **	<0.001	134
Total environmental modification score	Post-modification community environment	0.435 **	<0.001	137

Note. Spearman’s rank correlation coefficient (*ρ*) was used to assess the relationship between the variables, with ** *p* < 0.01.

**Table 2 nursrep-15-00400-t002:** Effects of Background Factors of Environmental Modifications on the Post-modification Care Environment.

			n = 311
Independent Variable: Background Factors of Environmental Modifications	Standardized Coefficient *β*	*p*-Value	Adjusted *R*^2^
Availability of desired services	−0.047	0.686	0.169
Service accessibility	0.286	0.020 *	
Information about necessary services	−0.166	0.162	
Presence of relatives, acquaintances, or friends to communicate with	0.090	0.452	
Presence of relatives, acquaintances, or friends to ask for help	−0.027	0.845	
Presence of relatives, acquaintances, or friends to share private matters with	0.147	0.292	
Relationships with professionals	0.095	0.356	

Note. Dependent variable = Total care environment score after environmental modifications. Results are based on multiple regression analysis; * *p*  <  0.05.

## Data Availability

Data are contained within the article and its Appendix A.

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
