# Peer review of "Associations Between Care Environments and Environmental Modifications in the Daily Living Settings of Children with Medical Complexity"

_nursrep, 2025, doi:10.3390/nursrep15110400_

Round 1

Reviewer 1 Report

Comments and Suggestions for Authors

Thank you for sending it to me so that I can evaluate this work.The study is well-designed and represents a study with significant findings.However, some revisions should be made.
The positions of the first and second sentences in the Introduction section should be swapped.
A table should be presented showing the primary diseases of the CMC children included in the study. Those with neurological disorders should also be indicated. CMC patients should generally be classified according to the primary disease.
How did you evaluate the given p-value in the statistical analysis? Under what conditions is the p-value considered significant?
Move the ethics committee approval sentence to Material and Methods 2.2 and adjust the numbering of the other sections accordingly.
Show the web-based survey mentioned in the study in parentheses.
In the power analysis of the study, you mentioned that the sample size should be 118, but only 26 families were included in the study? This study is not suitable for the required power.
Also, the number of families who responded to the survey was found to be 113. These numbers are confusing and inconsistent, and in no way align with the power analysis. First, define what the sample size mentioned in the power analysis refers to. Is it the number of families, the number of children, or the number of survey questions?
I didn't understand anything from Table 7. Please redesign Table 7.
The first paragraph of the discussion section should be rewritten. In the first paragraph of the discussion section, only explain what the significant results you found mean. For example, in Table 8, the only significant result is service accessibility. Explain why this result might have been significant.

Author Response

We would like to express our sincere gratitude  to the reviewers for their constructive comments and suggestions.

Details of our responses and the corresponding revisions are provided in the attached file for your review.

Reviewer 2 Report

Comments and Suggestions for Authors

Dear authors,

The title adequately informs the reader about the topic addressed in the article and would facilitate its search in databases.

The abstract provides a clear and comprehensive overview of the study.

The introduction addresses the current challenges regarding home care for children with complex medical needs.

The objective is clearly defined and I consider it appropriate for the topic addressed in this research.

The methodology is sufficiently detailed to allow for replication of the study, although:

- To use the independent samples t-test, it should have been verified that the data follow a normal distribution, an aspect that is not indicated in this section.

- To compare the environmental modifications in the pre- and post-intervention questionnaires, the paired samples t-test should not have been used since these are ordinal qualitative variables. The Wilcoxon test should have been used.

- To analyze associations between environmental modification scores and post-modification care environment scores, Spearman's correlation coefficient should have been used, as these are ordinal qualitative variables.

Since these are ordinal qualitative variables, I believe ordinal logistic regression would be more appropriate than multiple linear regressions.

The results are presented in detail with tables that facilitate understanding, although Table 7 is confusing; I assume this is because the modified data have not been removed.

The discussion thoroughly analyzes the results and establishes relationships with previous studies. However, once the meaning of CMC is explained, this acronym should be used throughout the document and its meaning repeated each time it is defined.

A comprehensive analysis of the limitations is provided, helping the reader draw their own conclusions from the results.

The conclusions are consistent with the results and address the stated objective.

The references are numerous and mostly up-to-date.

Kind regards.

Author Response

(The authors gave the same response as above.)

Round 2

Reviewer 1 Report

Comments and Suggestions for Authors

The authors fulfilled their responsibilities, and the paper can be published in its final form.

Reviewer 2 Report

Comments and Suggestions for Authors

Dear authors,
I have no further comments or suggestions to make.
Kind regards.